# What do families want to improve in the management of paediatric febrile neutropenia during anti-cancer treatment? Report of a patient/public involvement group

Bob Phillips,[1,2] Sarita Depani,[3] Jess Morgan[1]

[1]Centre for Reviews and Dissemination, University of York, Leeds, UK
[2]Regional Department of Paediatric Haematology and Oncology, Leeds Childrens Hospital, Leeds, UK
[3]University of Birmingham, Birmigham, UK

**Correspondence to**
Dr Bob Phillips; bob.phillips@york.ac.uk

## ABSTRACT

**Background** This study reports how parents and young people who had an experience of febrile neutropenia (FN) improved the design of a trial to inform the management of this condition. Five parents, a young person who had completed treatment and three clinician-researchers contributed.

**Methods** The group was formed after an invitation on social media and met via video conference. Many participants were from an existing childhood-cancer parent-involvement group. The initial questions asked during discussion were about the importance of the topic, the views on the need for a trial, which important outcomes should be measured and the practical aspects which would make it easier or more difficult for people to take part in it. The conversation occurred for an entire afternoon, was audio and video recorded, transcribed, analysed and checked by those involved. The fifth parent added to this via email.

**Results** The group altered the trial structure by proposing randomising of each child to one of the two management methods through the whole of their anti-cancer treatment, rather than randomising the study sites or the child at each visit. They felt that even if people declined taking part in the study in the first weeks of diagnosis, their views might change and they should be allowed to consent later. They also proposed methods of collecting important patient and family data, enriching the medical information gained in the study. Active follow-up, negotiated for each individual family, was also suggested.

**Conclusion** Trials for improving the management of FN in children and young people who are undergoing anti-cancer treatments should consider individual-patient randomisation, collection of 'quality of life' and 'experience of care' aspects using digital and paper-based methods, engage families in shared decision-making about management options and ensure adequate supportive information is available and accessible to all patients regardless of background, geographical location or age.

## INTRODUCTION

In high-income countries, the treatment of malignancies in childhood is associated with 5-year survival rates in excess of 80%.[1]

### What is already known on this topic?

► Febrile neutropenia (FN) is a common complication of childhood cancer therapy and is disruptive and resource intensive.
► Trials for reduction in intensity of treatment for fFN have previously been challenging for parents to accept.
► Parent/patient and public involvement in trials has improved study designs, research comprehension and engagement materials.

### What this study hopes to add?

► Parent/patient and public involvement in a proposed trial for reducing antibiotic treatment for febrile neutropenia (FN) led to changes in fundamental aspects of trial design.
► Proposed outcome assessments were enhanced by experts with experience who described the burden of treatment and trial procedures for FN.
► Video conferencing for parent/patient and public was effective despite the participants not knowing each other well.

This is possible through the use of intensive, toxicity-inducing regimens, where one-third of deaths in this group are the result of complications of therapy rather than directly due to the disease.[2 3] The cancer treatment often produces acute complications requiring unplanned hospitalisation, disruption, distress and strain for the young persons and their families.[4] One such complication is the co-occurrence of fever in the presence of neutropenia; this combination heralds a possible overwhelming infection and is considered a medical emergency.[5] The absolute risk of death or requirement of intensive care in such episodes is low, approximately 3%.[6] The challenge for families

and healthcare professionals is to effectively treat each episode, with minimum exposure to antibiotics and disruption of family life.

Research into episodes of febrile neutropenia(FN), and subsequent clinical practice guidelines have emphasised the need to treat promptly, assess the risk of each episode and treat with antibiotics chosen to address individual and local resistance patterns.[5] The methods of risk assessment and discontinuation of antibiotic therapy are, however, precautionary and conservative, treating two-thirds of patients with broad spectrum antibiotics unnecessarily.[7] Studies have shown that biomarkers of infection/inflammation seem to predict the risk of serious infection and its resolution, but have not been used to guide management.[8] Further refinement of the approach to FN has been identified as a research priority.[5]

In analogous situations with critically ill or immunocompromised hosts, such as adult or neonatal intensive care units, traditional management is similar to that of FN, with the prompt use of antibiotics and their discontinuation when infection has been excluded. Procalcitonin-led guidelines have been shown to reduce exposure to antibiotics and potentially improve mortality rates.[9 10]

The need to improve the management of FN led to the development of a research proposal to use procalcitonin, which is tested on a blood sample, to assist antibiotic decision-making during episodes of FN. It was felt that the decision on how to conduct such a study (which outcomes are important to measure, how to measure them and the possible barriers and solutions for a trial) was best taken with the engagement of clinicians, academics, and parents and young people who had had direct experience of anti-cancer treatment in childhood. Previous work has shown how such involvement led to improved research focus, better interview questions and enhanced the skills of children and young people undertaking such work.[11]

This paper reports the findings of a patient/public involvement (PPI) group which researchers, parents and young people convened to design a study to investigate procalcitonin-assisted decision-making in the management of FN in children undergoing anti-cancer therapy.

## METHOD

A request was made on social media for parents and young people who had had experience of childhood cancer therapy to consider taking part in a group to discuss the proposed trial. Volunteers were recruited and after initially attempting a face-to-ace meeting to promote inclusiveness and working together, a video conference platform (Zoom) was used to overcome geographical barriers. The researchers, all clinical doctors with additional academic roles, met in one location and the public contributors took part from their own homes. One participant could not get the integrated audio working; so he joined the conversation via telephone and mute video. The discussion lasted 2 hours and 15 min.

---

**Box 1    Original trial design**

► Site-randomised trial (randomising by hospital) using cluster or step-wedge approach.
► Use of single "quality of life" questionnaire at discharge.
► Contact with patient for trial purposes to occur only while in-patient – not after discharge.
► Antibiotic decision-making on procalcitonin measurements and clinical judgement without family involvement.

---

The session was structured to introduce FN, the existing evidence for the proposed intervention and the rationale for a randomised feasibility study (see box 1 for the initial plan). The PPI group had knowledge of clinical studies, including trials, in children and young people with cancer. The discussion followed a series of questions about the experience of FN, its management, the perceived challenges with current approaches and how the study could be best organised to meet these.

The session was video and audio recorded. The entire meeting was transcribed after audio immersion and the content thematically studied. Elements of the conversation related specifically to the design and conduct of the study were developed into themes and sub-themes. Elements related to the management of FN were examined within a framework derived from the themes developed in a relevant PhD thesis.[12] Following the meeting, a summary was shared and agreed, and the full report from which this paper is derived was reviewed by the PPI group.

The costs of the group were small; transport costs and light refreshments only for the researchers and a small fee for the video conferencing platform. The platform and the technologies were already owned by the participants. The participants volunteered their time, in line with their voluntary involvement in similar charitable activities, and did not receive payment.

As this was a patient/public engagement group, no ethical review was required. This is consistent with the INVOLVE definition of public involvement in research as 'research being carried out 'with' or 'by' members of the public rather than 'to', 'about' or 'for' them',[13] Despite the lack of a formal requirement for research ethics committee oversight, an ethical approach to such work is necessary. Such an approach has been described,[14] and the key elements of a fair choice to partake in the work, appropriate training and support to understand the questions asked, making sure access was as equitable as possible and providing recognition for the work were all considered in this project.

## RESULTS
### Participants
Four volunteer parents were part of the UK-based Paediatric Oncology Reference Team (PORT) organisation which consists of parents of children and young people

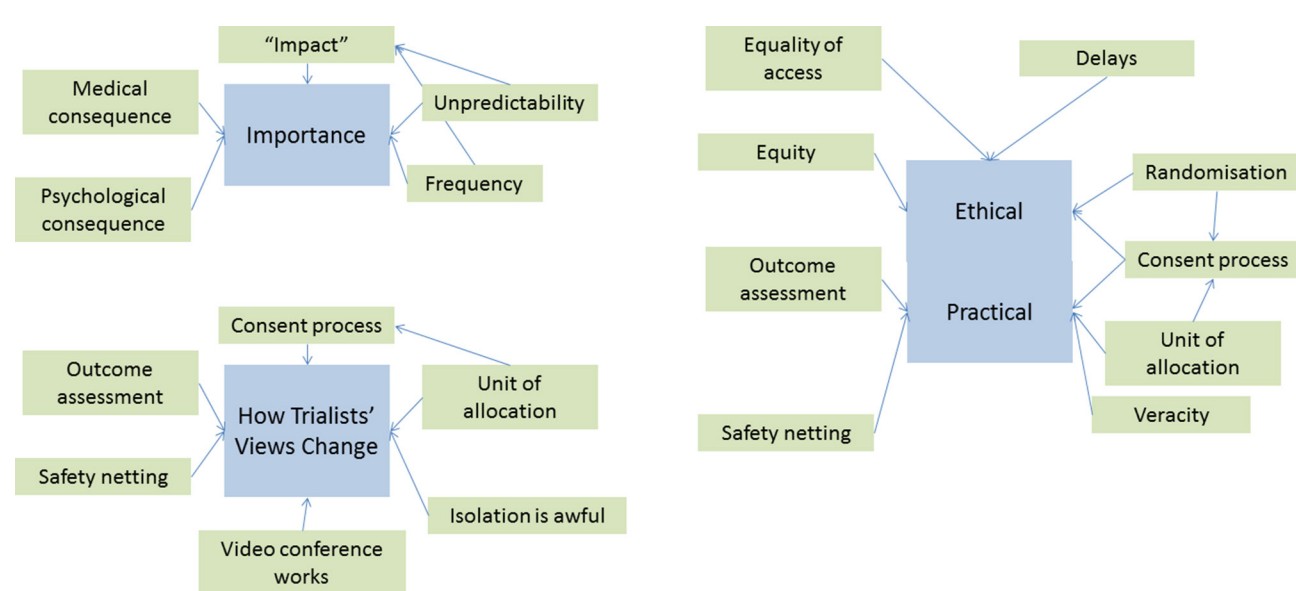

**Figure 1** Interaction of themes and sub-themes.

who have had experience of childhood cancer. Each was the mother of a child who had undergone cancer treatment, two with leukaemia and two with neuroblastoma. The children of three of these four parents had died of their disease. A patient who had had leukaemia as a teenager also took part in the group, though unrelated to the other participants. Of the three researchers, two were higher specialist trainees in paediatric oncology, and one a consultant, who was the only male member in this group. Each member of the group had prior experience with research in children's cancer beyond participation. The discussions involved descriptions of past experiences of admissions with fever and neutropenia. The experiences were from 2006–2017. During that time period, there has been a move to effect some reduction in the length of stay and marginally more consistency between centres in the UK.[15] Additional comments were added by a fifth PORT parent via email following the video conference.

### Study-specific themes

Information regarding the undertaking and conduct of the study was described under three major themes; 'importance', 'how PPI changes researchers' views' and 'practical and ethical'. The theme of 'importance' was formed by concepts of 'medical consequence', 'psychological consequence', 'impact', 'unpredictability' and 'frequency of FN'. (See figure 1).

*Importance:* The group members unanimously agreed that the management of episodes of FN was important because of its unpredictability, frequency and medical, psychological and social ('impact') consequences. They described particularly how variation in care across different hospitals was a source of concern to them and consistency would be a positive by-product of undertaking a trial:

we've got the 20-odd centres and pretty much everybody follows the same protocols [for anti-cancer treatment]… and I think it would be really reassuring for families if the POSCUs [Paediatric Oncology Shared Care Units] you know if we knew that everybody was doing the same thing [P1]

*'How PPI changes researchers' views'* describes the impact of this work: how PPI is important in modifying the initial, genuinely held presumptions and beliefs about the best ways to conduct such a trial for families and children and young people. These beliefs were drawn from the trial development group, the group of clinicians and researchers involved in designing the trial, which has over a century of experience in working with children's cancer in a variety of units and countries and specific expertise in studying supportive care in this group of patients. The impact of the group is seen in in the following aspects:

Allocation: The initial suggestion was for group randomisation, assigned arms to by clinical unit rather than by individual patients. However, the PPI input strongly steered towards individual randomisation, but with each individual receiving the same arm of management throughout the whole trial.

with paediatric oncology unlike many other things… everybody is on a trial… everybody… I mean… <snip>… I think everyone just expects that their treatment may be a bit different than everyone else's… [P2]

once you're randomised rather than each time coming in and by randomised each time, you're better off having 'this family is procalcitonin, this family is not' [P1]

Outcome assessment: The group members felt direct measures of patient experience would be important, more than 'quality of life' checklists. They suggested offering a daily experience journal of some form (paper or electronic, 'app' based), and believed this would supplement and enrich the medical data collected, such as admission duration, antibiotic duration and infective organisms.

> capturing that idea of burden beyond the hospital-based stuff and things that matter to the family, [P3]

They described how it would be important to measure the extra resources required because of FN admissions. The word 'impact' was felt to capture this rather than 'costs'.

> impact, because it's not just about extra costs, I mean if you're having to call in grandparents and you're having to call in favours left right and centre,<snip> I mean how many times can you ask the next door neighbours to collect your kids from school [P1]

Active safety netting: The researchers initially felt that the standard approach after the discharge of responsibilities passed to the family to 'return if unwell' would be safe and acceptable, but the PPI group thought an active approach to safety-netting was necessary, but should be individually negotiated.

> I think it needs to be more than just you phone up the hospital if you have any concerns. It should be either somebody coming around or phoning you and saying 'Do you have any concerns, do you have any concerns at all' and actually if you say yes …… giving you the option of coming back or having somebody over [P4]

Gaining consent: The researchers proposed, as with usual practice, the study would be offered once to families when the clinician believed it appropriate. The group members, with their experience of studies and information being exchanged, thought it would be fair to allow people to decline early on but have the opportunity to join the study if they changed their mind as their treatment journey progressed. They also floated the idea of families approaching clinicians to join rather than being invited.

> I don't think I even knew what febrile neutropenia was though, at the time when we were first giving our consent to all the other things? I think that's something that possibly comes with… further on… down the line… even a week or 2 weeks after you've given all those other consents. Because actually, the other consents are almost live-saving things… whereas this is a real choice… and I think that batching it in with

those initial forms of consents is almost taking away your flexibility of trying to consider it whether you want to do it or not [P4]

> You might also get people saying 'No' right at the beginning, if it's something they don't have to agree with, and then subsequently further on during their treatment when they can really see the how much of a headache that this can be… [giggles]

> - Do you think it's OK then to offer it twice? If somebody says no the first time? [R1]

> Yes – I think I do [P4]

> Because it's not like chemo A vs chemo B, it's not … it's not crucial like… you can opt in whenever you want [P2]

The theme of 'practical and ethical considerations' included the ethical aspects of; consent, randomisation, delays introduced by undergoing the trial, equity and equality, and the sharing of trial data. The practical aspects described outcome collection, safety netting, and ensuring the veracity of information collected in the trial.

Randomisation was considered a fair and ethical approach when in clinical equipoise. Along with this, a later discovery of one arm proving better than the other was not considered unethical; however if being on the study disadvantaged everyone (for example, if the treatment would be delayed while forms were completed) then it would not have been supported. A design which was accessible for the diversity of social, cultural and economic backgrounds of potential participants was essential. Confirmation of the scientific validity of the proposal and clinical equipoise was important to the PPI group. Prior systematic reviews with meta-analysis were felt to be a very comprehensive answer to this question.

Two of the three researchers have a strong interest in individual participant data meta-analysis. A question was asked about data sharing in this context and the PPI were very enthusiastic about being involved.

> Definitely share. I think the thing with paediatric oncology is that we do so many international trials together, because thankfully it is rare, but ultimately I think that… [snip] we're here ultimately to try to make things better for kids of the future and if that's part of it, and it is with these meta-analysis, then definitely. [P1]

The group members were concerned about study governance, for example ensuring the veracity of information collected during the study.

> are they [study groups] actually going to tell you the truth? [P2]

The members were keen to know if there would be some way of determining if the data collected were truthful and accurate; this seems to speak of a greater public awareness being required of the nature of health research governance within the country generally.

An offer of expression of interest in continuing to engage with the study governance was enthusiastically met with by the participants. One of the group members has joined the funding application as a co-applicant, has helped develop the grant application and will be involved in qualitative data collection and analysis as a co-investigator.

## Febrile neutropenia themes

Conversations in these discussions were mapped to the framework proposed by Morgan, developed to understand the decision-making processes involved in managing episodes of FN.[12] The overarching concepts she described were the quest for certainty, attaining mutual trust and the potential for realised discretion. These were all strongly endorsed in analysis of the group discussion.

The quest for certainty involves balancing the uncertainty of outcome of each episode of FN including an appreciation of probability, the use of protocols and guidelines to manage the risk and acknowledging the adverse elements of hospitalisation. The use of protective isolation, where the child and family are kept in a single room to avoid contracting infections from other hospitalised children, and source isolation, where the child is kept in a single room to prevent his/her infection from spreading to others, was viewed particularly negatively.

It was his cupboard—[child] called it his cupboard [P3]

Mutual trust had been a challenge, with the group describing individual healthcare practitioners in whom they did not place trust and the reciprocal of this, along with the negation of parental concerns.

the first time that I thought it was that they were taking too much precaution and I would have much preferred him to be at home taking tablets and things… monitored every so often… whereas the second time I think he needed more than what he was getting… and I think we were right both times actually [P4]

The ideal management of an episode of FN was one where safety was assured, hospitalisation was minimised, decisions discussed with families and support provided at home as desired by the family: the potential for realised discretion. The group readily acknowledged that decisions would need to be based on a range of factors, including home-to-hospital distance and the variability between parents and families in self-expressed confidence:

my sort of worry is that… the responsibility is even more on the parent as well… on top of like running the house… and its that sense of responsibility as well… like they're monitoring their child and being responsible for it… and like if something did happen would they feel guilty about it or not? [P5]

Exploring how the professionals were thinking about the episode, in terms of the likelihood of adverse outcomes and their considerations, was a strength in a shared decision-making approach which had been absent in many prior experiences.

I think it would be really helpful, [imitates Dr speaking]… we think its' like this [left] or we think it's like this [right]… and then chatting… and …… You know where you are coming from and where there is a difference and you know talking about … [P2]

seem to recall being in negotiations… situations where… ringing [PTC] consultants saying 'This is our situation… can you speak to them and and so on…' [P4]

## Reflections of clinical academics

The group discussions encouraged the three researchers to reflect on their previous approaches to FN and PPI in other studies. The more experienced researchers had participated in PPI before, but always on a face-to-face basis. The video conferencing allowed for a more diverse group of individuals to undertake the work, with the researchers in the same room on one screen facilitating. The makeshift arrangement of audio for one participant through phone served to reduce hierarchies, with collaborative suggestions and problem-solving forming an early 'win' for the group. This method worked well with the age and technological skills of this group, but may be less successful if a group with fewer technology skills or younger age were involved. The protocol changes suggested by the group were unexpected, as was the emphasis on the emotional burden of physical isolation. The researchers understood from this experience the value of listening to expert parents and young adults, and considering video or tele-conferencing to allow a greater number and range of people to take part in PPI events.

## DISCUSSION

The engagement of a group of parents and an ex-patient who had experience of cancer in children and young people, with researchers developing a study to improve the management of FN, led to changes in the proposed design of the trial and brought out a deeper understanding of the potential concerns of participants. The wider discussions about the nature of the experience of an episode of FN were congruent with prior work in the

---

**Box 2    Changes following consultation**

► Individual patient randomised trial (randomising by patient, not by episode).
► Consent permissible at any point during the cancer journey while still 'at risk'.
► Richer patient-experience measures – not just patient quality of life but family experience and their costs to be captured.
► Active follow-up after discharge.
► Explicitly encouraging shared decision making and sharing of results with families to decide antibiotic use.

---

field,[12] pointing particularly to actively involving parents and young people in sharing decisions about care.

The PPI altered how the trial would be structured, the randomisation of each child to one of the two management methods through the whole of his/her anti-cancer treatment, rather than randomising the study sites or the child at each visit (see box 2 for specific changes). The suggestion of providing multiple opportunities to be involved in the study was welcome and congruent with the description of an emerging expertise and empowerment of people through the childhood cancer journey.[16] They discussed practical methods of collecting data, which went beyond simple admission statistics and questionnaires, to enrich the information gained in the study. Active follow-up, with healthcare-initiated contact with the family but negotiated with each individual family, had not been originally considered by the researchers. The discussion also shed light on the experiences of the people involved in treatments of episodes of FN, with the ideal treatment being an individualised, negotiated approach with clear and safe guidance used consistently across all centres.

The expertise and prior relationships of the group members in similar situations may have enhanced the easy flow of ideas and conversations in this event. All members knew at least one other participant through in-person interactions in similar group settings or clinical interactions. Ice-breaking activities were extremely brief as there was little ice to be broken. None of the participants were paid for their time in undertaking this work. Though the INVOLVE guidelines suggest involvement should come with reimbursement, the group undertook the work as charity related to childhood cancer treatment, research and support and saw this as an extension of their other activities. Future PPI work with similar groups of people will benefit from considering holding group conversations via a video conferencing platform. The ready availability of web-cams and front-facing cameras on phones, tablet and laptops and the common use of video conversations at work and home mean these were acceptable methods for discussions with this group. There are limitations with this approach. It requires a familiarity and access to such equipment and access to a relatively stable internet connection. This may exclude particularly young people from disadvantaged backgrounds from PPI. It may also be very difficult to use for working with younger children or older family members, perhaps great-grandparents, who are unfamiliar with video conferencing. If the approach is used, it may be beneficial to have a 'test run' prior to the meeting to allow any technical challenges to be met; we would suggest a sensible way forward would be allocating a period of time for 'drop in' connections to confirm everything is working well. A backup, as simple as a telephone line, is also very helpful.

We used social media (Twitter) to recruit the participants; as all the researchers had prior experience of working with PORT, and 'tagged' them into a post, this may be considered a mixture of open and direct messaging. This type of use has been fairly widely undertaken previously[17] and has advantages and disadvantages. It carries little direct risk as it doesn't ask people to engage in a discussion in a forum (such as Facebook or Blog comments), but its reach is limited to those who already follow one of the accounts which posts or re-tweets the invitations. It provided an excellent opportunity to draw in active PPI parent volunteers, but did not attract a large number of young people. Directly advertising the opportunity to be involved in the work to young people via other groups, such as Young People's Advisory Groups hosted by organisations such as the National Cancer Research Institute or Clic-Sergant, or advertising through the Teenage Cancer Trust, may have resulted in more than one young person getting involved.

The findings of this study have immediately influenced an application for a feasibility study of procalcitonin-guided management of FN. They will also influence the ongoing development of clinical practice by dissemination through children's and young people's professional network groups. The participants in this group have expressed a wish to be a part of the steering committee of a trial addressing this issue, and the ongoing study development will also seek further involvement of young people, following INVOLVE guidelines.[17 18] One of the group members has joined the study as a co-applicant, developing the grant and plans to be involved as a co-investigator.

**Contributors** This study was conceived by BP and SD, and developed with the assistance of JM. The audio was transcribed and analysed by BP initially with input from JM and SD. BP drafted the paper, and was critically revised and developed by JM and SD. The PPI group read and agreed with the content of the paper. The authors very gratefully acknowledge their input into this specific work.

**Funding** This research received no specific grant from any funding agency in the public, commercial or not-for-profit sectors. BP was supported by an NIHR Post-doctoral fellowship: grant number PDF2014-10872. The views expressed in this publication are those of the authors and not necessarily those of the NHS, the National Institute for Health Research or the Department of Health and Social Care (DHCS).

**Competing interests** None declared.

**Patient consent for publication** Not required.

**Provenance and peer review** Not commissioned; externally peer reviewed.

purpose, provided the original work is properly cited, a link to the licence is given, and indication of whether changes were made. See: https://creativecommons.org/licenses/by/4.0/.

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
