## [Reviewer comments · BMJ Paediatrics Open]

ARTICLE DETAILS

TITLE (PROVISIONAL)	What do families want to improve in the management of paediatric febrile neutropenia during anti-cancer treatment? Report of a patient/public involvement group.
AUTHORS	Phillips, Bob; Depani, Sarita; Morgan, Jess

VERSION 1 – REVIEW

REVIEWER	Reviewer name: Derek C Stewart Institution and Country: Patient Advocate, United Kingdom Competing interests: Active in promoting patient involvement in all aspects of research as a means of enhancing studies in terms of need, value and relevance
REVIEW RETURNED	31-Oct-2018

GENERAL COMMENTS	Please note: I have used some of the terms (in brackets) from the National PPI Standards that the authors may wish to consider referencing at times in the document or in future papers. This work also reflects Kristina Staley's report that talks about PPI making a difference to 'research' and 'people'. Comments: This is extremely timely contribution to the dialogue around patient/public involvement (PPI) in research. It provides a constructive illustration of the practical value of such involvement, offers helpful advice about the process as well as describing the difference this made (Impact). The article demonstrates the merit of having a proper plan about how you are going to involve patients and parents: the way you are going to find people; the need for a structure for the discussions; how to capture the contributions; and to describe the differences and impact these have made. This type of structural clarity is missing from much of the writing about PPI. First of all, the reach was by social media rather than handpicking a few select individuals to sit on a committee or attend a discussion group. Although many of the respondents were known to the researchers, this openness gave opportunities for others to join and probably helped raise awareness. This shows the extent to finding different means of PPI (Inclusiveness and Working Together). Using video conferencing means that people can fit involvement into their lives rather than always travel to a meeting. The fact that the team had a contingencies to enable a telephone conference element and contribution by email can be viewed as inclusive and helpful communication. The structure for the discussion of Importance, Value, Outcomes and Advice could be outlined a little stronger in the article as it offers a useful framework for other to follow and adapt.
--

	Similarly the clustering of the issues into study specific themes are again helpful guides. The real power of this article lies in the examples of how this involvement changed the actual design of the study in a fundamental manner - ie the randomise method (Impact). The clarity with which the group recommended the use of clinical data was also interesting as it is an issue worrying many researchers. The recording and reporting (Communications and Impact) of the dialogue between the team and those affected by paediatric febrile neutropenia is well composed offering a mix of quotes, comment and analysis. The description of being 'in a cupboard' perfectly captures the line that this exercise brought about a 'deeper understanding'. Finally, I like the fact that the group have shown a willingness to continue to be involved and that they will be included on the Trial Steering Group (Governance). The authors should be encouraged to continue writing about how the patient voice is helping to shape and influence this work. It will be interesting to see how both the patients and team are supported and what they feel they have learned from the experience over time (Support and Learning).
--	---

REVIEWER	Reviewer name: Louca-Mai Brady Institution and Country: Kingston University, UK Competing interests: I sit on a study steering committee with Bob Philips.
REVIEW RETURNED	07-Nov-2018

GENERAL COMMENTS	This is a really interesting article which I think could make a worthwhile contribution to the literature on patient and public involvement in health and social care as well as paediatric oncology. As the former is my area of expertise rather than the latter the following comments are focused on this submission as an article on PPI. I think the article needs to locate this work in relation to the PPI literature and particularly the literature on children and young people's involvement e.g. why the research team wanted to do PPI, evidence on benefits and good practice. Some recent references are included below which may be useful + I would suggest following the GRIPP2 guidance as this is now the accepted standard on reporting PPI. The article reports clearly how the public involvement group was set up and run but I think does so fairly uncritically. Some reflection on the cons as well as the pros of the approach taken and emerging learning would be helpful. Is there anything the authors think could have been done better or differently regarding who was involved and how? E.g was the young person who attended a child of one of the adult contributors and if so what were the implications of this? Why was only one young person involved? Were there advantages as well as disadvantages of having adults and young people meeting together? Would the authors have liked to have ongoing involvement/more than one meeting with the group and if so what were the barriers to this?
---

In the abstract and generally: as well as how the PPI group informed the trial plans - are there any general lessons on PPI with parents and young people in the design stage of research and generally? Language is inconsistent for those involved: public contributors are referred to variously as 'volunteers', 'the PPI group, and 'the group' & the research team as 'the trialists', 'trial group' 'clinical trialists' and 'clinical academics'. PPI group members/public contributors and researchers might be simpler.

Key messages: 'What is known': PPI has done a lot more than modify designs and information leaflets (ref: point about drawing on literature above. 'What this study adds': add a point on how this article adds to the literature on PPI/PPI with families and CYP
Methods: as above more discussion on the pros and cons of, for example, involving parents and young people in the same group + video as opposed to face to face meetings. Much of the existing evidence on young people's involvement seems to say that face-to-face is better so I'm interested in why the authors found this not to be the case with this project and whether they this approach might potentially exclude some, while enabling the involvement of others? The authors say that 'the participants volunteered their time and were not paid' but NIHR INVOLVE guidance is that public contributors should be paid for their time. This should probably be mentioned or briefly discussed.

'A patient/public engagement group of experts through experience in the development of a study' - could this wording be clarified/simplified? Also in this paragraph explain why ethical review wasn't required for readers who may be unfamiliar with the NIHR INVOLVE/HTA guidance on this.

'Study-specific themes': I agree that 'PPI is important in modifying....presumptions and beliefs' and this would be a good point to draw in some of the literature. Similarly where the authors state that 'this seems to speak of a greater public awareness being required'.

I was interested in knowing more about plans for PPI when the trial is underway, if and how the group informed these plans and any ongoing involvement in the study (e.g. as coapplicants). This is only mentioned in passing in the final sentence but I think merits further discussion.

Example refs/resources:

CYP's involvement

Bird D., Culley L., and Lakhanpaul M. (2013) Why collaborate with children in health research?: an analysis of the risks and benefits of collaboration with children. Archives of Disease in Childhood: education and practice edition. 98 (2), pp. 42-8.

Brady, L.M., Templeton, L., Toner, P., Watson, J., Evans, D., Percy-Smith, B., and Copello, A. (2018). Involving young people in drug and alcohol research. Drugs and Alcohol Today, 18 (1), pp.28-38. <https://doi.org/10.1108/DAT-08-2017-0039>

Fleming, J., & Boeke, T., eds. (2012) Involving children and young people in health and social care research. London: Routledge.

Dennehy R, Cronin M, Arensman E. Involving young people in cyberbullying research: The implementation and evaluation of a rights-based approach. Health Expectations. 2018;00:1–11.

<https://doi.org/10.1111/hex.12830>

	INVOLVE (2014) Guidance on the use of social media to actively involve people in research. Southampton: INVOLVE [online]. Available from: http://www.invo.org.uk/posttypepublication/guidance-on-the-use-of-social-media/ INVOLVE (2016) Involving children and young people in research: top tips for researchers. Southampton: INVOLVE [online]. Available from: http://www.invo.org.uk/posttypenews/involving-children-and-young-people-in-research-top-tips-and-key-issues/ INVOLVE (2016b) Reward and recognition for children and young people involved in research – things to consider. Southampton: INVOLVE [online]. Available from: http://www.invo.org.uk/wp-content/uploads/2016/05/CYP-reward-and-recognition-Final-April2016.pdf Parsons, S., Wendy Thomson, W., Cresswell, K., Starling, B., McDonagh, J.E. (2018) What do young people with rheumatic conditions in the UK think about research involvement? A qualitative study. Pediatric Rheumatology, 16(1):35. https://www.ncbi.nlm.nih.gov/pubmed/29793489 Mitchell SJ, Slowther A, Coad J, et al (2018) Ethics and patient and public involvement with children and young people. Archives of Disease in Childhood - Education and Practice. https://ep.bmj.com/content/early/2018/02/08/archdischild-2017-313480 Staniszewska, S., Brett, J., Simer, I., Seers, K., Mockford, C., Goodlad, S., Altman, D.G., Moher, D., Barber, R., Denegri, S., Entwistle, A., Littlejohns, P., Morris, C., Suleman, R., Thomas, V. and Tysall, C. (2017), "GRIPP2 reporting checklists: tools to improve reporting of patient and public involvement in research", Research Involvement and Engagement 3 (13) https://researchinvolvement.biomedcentral.com/articles/10.1186/s40900-017-0062-2 PPI standards 4Pi National Involvement Standards: https://www.nsun.org.uk/faqs/4pi-national-involvement-standards NIHR national PPI standards: http://www.invo.org.uk/posttypepublication/national-standards-for-public-involvement/
--	---

VERSION 1 – AUTHOR RESPONSE

Reviewer: 1

Reviewer name: Derek C Stewart

Reviewer: 2

Reviewer name: Louca-Mai Brady

Reviewer: 1

Institution and Country: Patient Advocate

United Kingdom

Reviewer: 2

Institution and Country: Kingston University, UK

Briefly, the main elements are, in response to Reviewer 1:

Comments to the Author

Please note: I have used some of the terms (in brackets) from the National PPI Standards that the authors may wish to consider referencing at times in the document or in future papers. This work also reflects Kristina Staley's report that talks about PPI making a difference to 'research' and 'people'.

Response summary:

We would like to thank the reviewer for these very positive and constructive comments. We've adapted our manuscript in response to these, and those of the other reviewer, to use more acknowledged terminology to enhance clear communication. We are particularly pleased our work was accessible to someone from outside the 'niche' world of paediatric oncology.

The elements raised by Reviewer 2:

Comments to the Author

This is a really interesting article which I think could make a worthwhile contribution to the literature on patient and public involvement in health and social care as well as paediatric oncology. As the former is my area of expertise rather than the latter the following comments are focused on this submission as an article on PPI

I think the article needs to locate this work in relation to the PPI literature and particularly the literature on children and young people's involvement e.g. why the research team wanted to do PPI, evidence on benefits and good practice. Some reflection on the cons as well as the pros of the approach taken and emerging learning would be helpful. how the PPI group informed the trial plans Language is inconsistent for those involved

Response summary:

We would like to greatly thank the reviewer for their work in improving our article, providing a great selection of evidence to deepen our work and thoroughly appreciate the work we are attempting to do. We have described more critically our method, highlighting the weaknesses and suggested reasons for success, developed the background and where this advances our knowledge of PPI further, and put in a more consistent approach to 'naming' as well as expanding what actually occurred in the information sharing part of the group discussion.

VERSION 2 – REVIEW

REVIEWER	Reviewer name: Derek Stewart Institution and Country: Freelance Patient Advocate, formerly with the NIHR Clinical Research Network Competing interests: None
REVIEW RETURNED	07-Dec-2018

GENERAL COMMENTS	Derek Stewart
---------------

REVIEWER	Reviewer name: Louca-Mai Brady Institution and Country: Kingston University, UK
-----------------	--

	Competing interests: I sit on a study steering committee with Bob Philips.
REVIEW RETURNED	21-Dec-2018

GENERAL COMMENTS	I think this paper is much improved and many of my original points have been addressed. But on the subject of video as opposed to face to face meetings this approach might potentially exclude others than just 'younger children or those unfamiliar with video conferencing' - e.g. some disabled CYP and/or CYP from disadvantaged backgrounds to do not have ready access to technology or the internet. My point really is that technology and can make involvement more accessible and inclusive to some but potentially exclude others. Perhaps add a sentence on this and ref the INVOLVE guidelines on social media and PPI?
--

VERSION 2 – AUTHOR RESPONSE

In response to Reviewer 2 comment:

I think this paper is much improved and many of my original points have been addressed. But on the subject of video as opposed to face to face meetings this approach might potentially exclude others than just 'younger children or those unfamiliar with video conferencing' - e.g. some disabled CYP and/or CYP from disadvantaged backgrounds to do not have ready access to technology or the internet. My point really is that technology and can make involvement more accessible and inclusive to some but potentially exclude others. Perhaps add a sentence on this and ref the INVOLVE guidelines on social media and PPI

We are happy to expand our rather condensed sentence on this point, and emphasise where we have quoted the INVOLVE social media guidelines (ref 17). We've also taken the opportunity to be a bit clearer on the benefits and limitations of the social media approach we used.

“The ready availability of web-cams and front-facing cameras on phones, tablet and laptop computers, and the common use of video conversations in work and home life mean these were acceptable methods to have discussions with this group. There are limitations with this approach. It requires a familiarity and access to such equipment, and access to a relatively stable internet connection. This may exclude PPI, particularly young people, from disadvantages backgrounds. It may also be very difficult to use to work with younger children, or older family members, perhaps great-grandparents, who are unfamiliar with video conferencing.”

And then

“We used social media (Twitter) to recruit the participants; as the researchers all had prior experience of working with PORT, and ‘tagged’ them into a post, this may be considered a mixture of open and direct messaging. This type of use has been fairly widely undertaken previously 17 and has advantages and disadvantages. It carries little direct risk, as it doesn’t ask for people to engage in discussion in a forum (such as Facebook or Blog comments), but its reach is limited to those who already follow one of the accounts which post, or re-tweet, the invitations. It provided an excellent opportunity to draw in active PPI parent volunteers, but did not attract a large number of young people. Direct advertising of ..”

We have added the Box titles as suggested by the Editor In Chief and have deleted ‘sites randomised’ as this is redundant, given we’ve just described site-based randomisation in the line above.

We hope this now meets your approval and hope to be published soon.